# A Motion Deblurring Network for Enhancing UAV Image Quality in Bridge Inspection

Jin-Hwan Lee [1], Gi-Hun Gwon [1], In-Ho Kim [2,*] and Hyung-Jo Jung [1,*]

[1] Department of Civil and Environmental Engineering, Korea Advanced Institute of Science and Technology, Daejeon 34141, Republic of Korea; archi_tensai@kaist.ac.kr (J.-H.L.); irlgns@kaist.ac.kr (G.-H.G.)
[2] Department of Civil Engineering, Kunsan National University, Gunsan 54150, Republic of Korea
* Correspondence: inho.kim@kunsan.ac.kr (I.-H.K); hjung@kaist.ac.kr (H.-J.J)

**Abstract:** Unmanned aerial vehicles (UAVs) have been increasingly utilized for facility safety inspections due to their superior safety, cost effectiveness, and inspection accuracy compared to traditional manpower-based methods. High-resolution images captured by UAVs directly contribute to identifying and quantifying structural defects on facility exteriors, making image quality a critical factor in achieving accurate results. However, motion blur induced by external factors such as vibration, low light conditions, and wind during UAV operation significantly degrades image quality, leading to inaccurate defect detection and quantification. To address this issue, this research proposes a deblurring network using a Generative Adversarial Network (GAN) to eliminate the motion blur effect in UAV images. The GAN-based motion deblur network represents an image inpainting method that leverages generative models to correct blurry artifacts, thereby generating clear images. Unlike previous studies, this proposed approach incorporates deblur and blur learning modules to realistically generate blur images required for training the generative models. The UAV images processed using the motion deblur network are evaluated using a quality assessment method based on local blur map and other well-known image quality assessment (IQA) metrics. Moreover, in the experiment of crack detection utilizing the object detection system, improved detection results are observed when using enhanced images. Overall, this research contributes to improving the quality and accuracy of facility safety inspections conducted with UAV-based inspections by effectively addressing the challenges associated with motion blur effects in UAV-captured images.

**Keywords:** UAV inspection; motion deblurring; image quality enhancement; generative adversarial network; object detection

## 1. Introduction

The utilization of unmanned aerial vehicles (UAVs) for facility safety inspections has brought about a significant transformation in various industries, including civil engineering. It offers cost effectiveness and increased operational efficiency compared to conventional human-based safety inspections. Structural monitoring, which relies on reliable technological methods to assess infrastructure conditions, is a crucial practice to ensure the long-term serviceability of targeted structures. Manual inspection conducted by human worker, a widely employed approach for monitoring structures over the years, entails evaluating the condition of a structure based on subjective judgment. However, this method presents several challenges, including exposing inspectors to hazardous environments, consuming substantial time and financial resources, and yielding results that may not be fully dependable.

In recent years, a promising solution to overcome these limitations has emerged through the active utilization of unmanned aerial vehicles (UAVs) for structure monitoring. This method involves employing UAVs equipped with high-resolution vision sensors or cameras to capture detailed images of a structure's exterior. The UAV can be guided along a predefined flight path or manually controlled to conduct the monitoring operation

effectively. Based on these characteristics and advantages, numerous studies have been conducted regarding the utilization of images captured by unmanned aerial vehicles (UAVs) for bridge inspection [1–4]. Depending on the type of camera equipped on UAVs, research has been conducted to detect various types of damages, such as crack detection using RGB images and the quantification of deterioration, like concrete spalling and delamination, using thermal images [5,6]. Furthermore, research has proposed comprehensive frameworks from the pre-inspection to post-inspection phases of bridge assessment [7,8]. However, despite these efforts, challenges remain in addressing the quality issues of images captured in dynamic environments. Particularly, issues related to image quality degradation due to environmental factors such as wind or low lighting have been highlighted as hurdles that UAV-based bridge inspection technologies need to overcome. Among various quality degradation phenomena like noise, blur, low lighting, and defocusing, motion blurring is a problem that is difficult to overcome through post-processing. Especially when capturing large-scale structures like bridges, there is a tendency to choose the shortest path to minimize operational time, which can be constrained by time limitations. As a result, problems stemming from environmental factors like vibrations or wind affecting rapidly moving UAVs have been identified as issues directly impacting image quality. In addressing the motion blur problem, some researchers have focused on the detection and removal of blurry images. To identify areas of blurriness within an image, prior methodologies predominantly focused on assessing the sharpness of image edges [9] or calculated the gradient magnitude [10]. Alternatively, Su et al. [11] adopted a distinctive approach for blur detection by combining several localized characteristics, including the power spectrum slope, Gradient Histogram Span, and Maximum Saturation, all of which become apparent in the presence of blur. This approach also made a substantial contribution to addressing the problem of image restoration by classifying blur into two distinct types, motion blur and out-of-focus blur, relying on image patches as the basis for categorization. Another method proposed by Bang et al. [12] entails the comparison of blur metric values derived from adjacent frames through the application of moving averages. However, it is important to note that these techniques continue to depend on threshold settings, which can be problematic to ascertain with precision. In addition to blur area detection, research on deblurring techniques aimed at directly improving the quality of blurry images has also been actively conducted. Most non-uniform deblurring methodologies initiate their process with the foundational assumption that the observed blurred image ($B$) results from the convolution of an underlying sharp image ($I$) with a blur kernel ($K$), which is determined based on a motion field.

The family of image deblurring approaches can be classified into blind and non-blind deconvolution methods. Non-blind deconvolution assumes prior knowledge of the blur kernels present in an image, while blind deconvolution is conducted without any additional information on the blur kernels. Early work predominantly focused on non-blind deconvolution methods, often relying on algorithms such as Richardson–Lucy, Wiener filter, or Tikhonov filter to perform deblurring [13,14]. In more recent times, there has been the development of blind deconvolution approaches aimed at handling situations in which the blur kernel remains unidentified. Gupta et al. [15] estimated the spatially non-uniform blur kernel resulting from camera vibration and deconvolved the image using a motion density function. Nonetheless, it is evident that there is potential for further refinement in the estimation of the blur kernel. Tai et al. [16] focused on spatially varying camera motion blur and proposed a projective motion deblurring model based on the Richardson–Lucy algorithm. However, it is worth noting that their approach necessitates knowledge of a pre-defined camera motion path, thereby constraining its practical applicability. Sieberth et al. [17] introduced two deblurring methodologies, one based on the Fourier approach and the other utilizing the edge-shifting technique. While both methods yielded outstanding results in deblurring aerial images, they exhibited certain limitations related to the requirement of precise transformation parameters. Additionally, the edge-shifting approach faces challenges in detecting complex crack patterns. The non-

uniform blind deblurring algorithms mentioned earlier showcase proficient image deblurring capabilities. However, their efficacy is hinged on a multitude of problem-specific parameters and configurations, encompassing internal camera parameters, external motion functions, thresholds, and termination criteria. Consequently, these algorithms encounter challenges when it comes to practical implementation and generalization in real-world scenarios. In the early stages, an analysis was conducted on the impact of motion blur in UAV-based images on feature matching using SURF and brute force matching [17]. The results revealed that even minor displacements of the camera, leading to image blur could have a significant adverse effect on image processing. Furthermore, through related studies, it became evident that the quality of images captured by UAVs directly influences the outcomes of visual inspections on structures [8]. They demonstrated that excluding blurry images prior to photogrammetric processing could greatly enhance feature detection and reconstruction. Additionally, they observed that the extent of motion blur caused by camera shake led to a reduction in image sharpness and a decrease in the accuracy of crack detection.

In recent times, owing to the progress in deep learning technology, there has been a significant focus on investigating learning-based deblurring techniques. One such approach involves the utilization of convolutional neural networks (CNNs) for the estimation of the blur kernel function. Extensive research has been conducted to develop a convolutional neural network (CNN) capable of predicting blur kernels at the patch level to effectively eliminate non-uniform blur in images [18]. Furthermore, considerable attention has been directed towards research that explores the utilization of convolutional neural networks (FCNs) for image deblurring through the estimation of motion flow [19]. Another approach is using multi-scale CNNs to deblur images without explicitly estimating the blur kernels [20,21]. Similarly, Generative Adversarial Networks (GANs), particularly the DeblurGAN model [22,23], have shown promising results in image deblurring with reduced computation time, without relying on explicit blur kernel estimation. In the context of UAV-based crack images, they present an interesting case for analysis. Due to the hairline nature of many cracks, they can be easily distorted by blurring, leading to decreased accuracy in crack detection. Nevertheless, the previously mentioned deblurring techniques have not undergone dedicated testing on UAV-based images of cracks, leaving room for potential enhancements in this domain.

Similarly, Generative Adversarial Networks (GANs), particularly the DeblurGAN model [22,23], have shown promising results in image deblurring with reduced computation time, without relying on explicit blur kernel estimation. In the context of UAV-based crack images, they present an interesting case for analysis. Due to the hairline nature of many cracks, they can be easily distorted by blurring, leading to decreased accuracy in crack detection. Liu et al. [24] conducted a study aimed at removing blur from crack images captured by UAVs using a deblur GAN model. Given the challenges in obtaining corresponding image pairs in real-world scenarios, artificially generated blurry images through motion blur simulation were employed. Notably, this research focused on the domain of crack images, utilizing an existing deblur GAN model as the generator network and the VGG16 network [25] as the discriminator network. This novel approach, distinct from previous studies, yielded impressive deblurring results, representing significant advancements in crack identification. Nevertheless, it is important to note that the use of artificially created blurry images through motion blur simulation and clear images from the same frame as data pairs has limitations in capturing the comprehensive characteristics of real-world blurring.

In this study, the main goal is to minimize blurring between bridge monitoring using UAVs. Basically, the characteristics of blur in images taken in a static state and the characteristics of motion blur in images acquired from UAVs are different in terms of the shape, size, and shape of the blur kernel mentioned earlier. Technological solutions, such as UAV speed control, camera vibration control, and sufficient illumination, exist for the suppression of motion blur affected by multiplicative artifacts. However, more effective solutions are required, as motion blur can occur within the image during filming due to the

effects of flight time constraints and instability due to the UAV battery limit, flight path deviation due to GPS signal shaded area, and error in shooting angle.

Therefore, in this study, GAN-based image deblurring networks and UAV image domains are utilized to differentiate them from existing studies by solving problems through image post-processing rather than a hardware approach. Typically, to employ GAN for image deblurring, a dataset consisting of paired sharp and blurred images is required [26]. However, in the case of images captured by UAVs, reference images are often unavailable, necessitating the artificial synthesis of blurred images by combining consecutive frames [27]. Nevertheless, this process may introduce discrepancies between artificially synthesized blur characteristics and those occurring naturally. To address this, this paper verifies the applicability in the UAV image domain by connecting a module that learns deblurring with a module that learns blurring characteristics.

In other words, this paper contributes in the following ways: Firstly, based on the recognition of domain differences between artificially synthesized blurry images and actually captured blurry images, it generates synthesized blurry images that closely resemble real-world blurry images. Secondly, it trains the model using the synthesized blurry images, which closely resemble real ones and actual sharp images as data pairs. Thirdly, it employs the trained GAN model to remove blur in UAV images used for bridge inspection and validates its effectiveness using image quality metrics and a deep learning model for object detection.

## 2. Proposed Motion Deblurring Network

### 2.1. Challenging Issues about Motion Blur in Bridge Inspection Using UAV

The drone image-based bridge monitoring process is largely divided into pre-inspection, inspection, and post-inspection phases [7]. Bridge damage assessment uses image processing techniques to identify, classify, and quantify damage assessment criteria using images acquired in the inspection phase. However, if motion-blurred images are used, a proper bridge damage assessment cannot be performed and, if necessary, a re-shooting of the area must be performed. Additional costs and labor are inevitable and can be a major obstacle to quick inspections of large structures, such as bridges. However, if the process of deblurring within the post-inspection phase is carried out, it can be very useful not only to identify damage but also to create inspection maps or 3D models of bridges required in the process of identifying damage. Therefore, the challenges addressed in this study are essential parts of enhancing the completeness of UAV-based bridge monitoring technology.

Motion blur is a challenging and ill-posed problem that often arises during image acquisition. Undesirable image blur degradation can be caused by camera vibrations and high-speed object motion, which are the two primary sources of motion blur commonly encountered in the image acquisition process. With the advancement and growing accessibility of UAV technology, unmanned aerial vehicles (UAVs) are progressively finding utility as monitoring instruments across diverse facilities. Typically, these inspection UAVs are equipped with an array of sensors, including a high-resolution vision system, to facilitate large-scale inspections. Nevertheless, variations in external conditions, such as weather fluctuations, wind-induced vibrations, or abrupt operator input errors, frequently give rise to motion blur, resulting in undue oscillation of the vision system as illustrated in Figure 1.

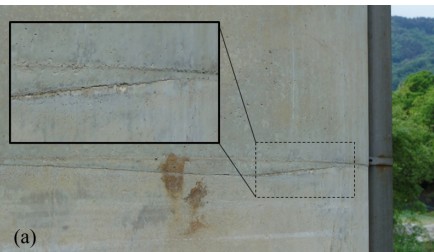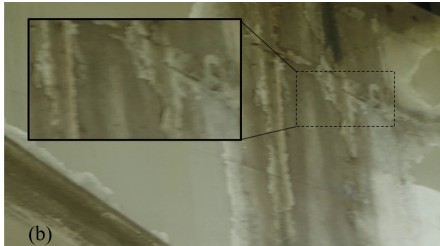

(a)  (b)

**Figure 1.** A case where motion blur occurred in the UAV image for bridge inspection. (**a**) Original image (**b**) Motion blurred image.

Since the introduction of GAN models by Ian Goodfellow [28], various studies have been conducted to improve the performance of GAN models. Deblur GAN [22], Cycle-Deblur GAN [29], and Reblur GAN [30] are prominent models for restoring blurred images. In general, in a single image deblurring, a blurred image $B$ is usually modeled as:

$$B = K * L + N \tag{1}$$

where $K$, $L$, and $N$ represent the blur kernel, the latent sharp image, and additive noise, such as white Gaussian noise, while $*$ denotes the 2D convolution operator. The objective in image deblurring is to estimate both $K$ and $L$ from the input blurred image. Nevertheless, this problem is inherently imprecise, primarily due to the existence of an infinite number of solutions capable of generating the same result denoted as $B$. Generally, a solution to this problem necessitates the availability of prior knowledge regarding $K$ and $L$, but the absence of such prior information may result in inaccuracies in the deblurring process. Consequently, a variety of diverse methodologies have been proposed over the past decades to address the inherent imprecision of this image deblurring problem. In particular, blind-deconvolution methods that perform deblurring are often applied to UAV images without prior information on blur kernels. This approach is often assumed to apply a spatially uniform blur kernel to the entire blur image. However, there is a limit to the uniform blur kernel-based method because multiple factors such as camera rotation changes or object movements work in the elements that generate actual motion blur. In addition, despite many prior studies for non-uniform blur kernel estimation to compensate for these limitations, it did not completely suppress dynamic blur (motion blur) caused by object movement or camera movement. Therefore, to address this, we perform blur and deblurring learning using a generative adversarial model, one of the image-to-image translation methods, rather than a step-by-step kernel estimation method.

### 2.2. The Proposed Motion Deblurring Network Based on Improved Standard GAN

#### 2.2.1. Architecture

The existing kernel estimation methods for image deblurring have limitations in accurately resolving the blur problem in UAV images captured in dynamic environments. To overcome this challenge, this study proposes a new deblurring model based on GAN with the ultimate goal of removing the blurring effect from UAV images.

While there are several deblurring techniques based on CNN, they are not end-to-end methods and exhibit low efficiency, with processing times exceeding 10 s per image. In contrast, GAN networks offer the advantages of fast processing times for individual images and accurate and effective blur removal. Generally, GAN-based deblurring methods involve pairing artificially synthesized blurred images with corresponding clear images and operate by generating sharp images from the blurred counterparts. However, since the domain characteristics of artificially synthesized blur images differ from those of real-world blur images, accurate deblurring effects cannot be reliably achieved. Therefore, the motion deblurring network that is newly constructed, based on the GAN model, includes a blur generation module that generates blur images closely resembling real-world blurs from the dataset construction stage. It also incorporates a deblurring module that performs the final deblurring process. By combining these modules, the proposed method aims to achieve more accurate and effective deblurring results for UAV images. The deblur network consists of two main modules: a deblur module and a blur module. Similar to previous image deblurring works, the deblur module is trained on paired sharp and blurry images to recover sharp images from blurry ones. These paired images are obtained from the blur module, which is trained on unpaired data, using sharp images and blurry images captured by UAV in bridge inspection. The overall architecture of the proposed framework is illustrated as shown in Figure 2. In Figure 2 below, block G means the generator network and block D means the discriminator network.

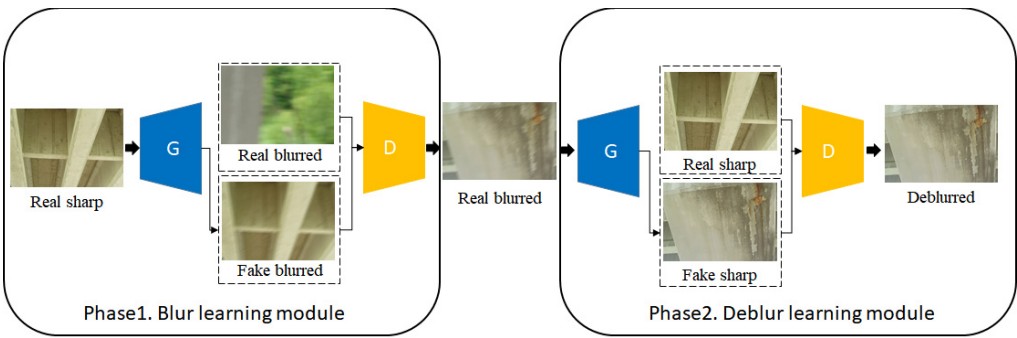

**Figure 2.** The architecture of the proposed motion deblurring network.

This model utilizes the concept of relative loss to enhance the standard GAN model. In contrast to deblurring models employing traditional GANs, the discriminator and generator in this approach are not trained solely to increase the probability of input data being considered real and the generated data appearing authentic. Rather, their objective is to evaluate whether a given real image is superior to the generated image in realistically portraying blurriness. During the training phase, clear images are provided as input to the blur learning module, and the outcomes are subsequently fed into the deblur learning module for training purposes. Both modules' generators produce corresponding images, while the discriminator generates a more refined composite image. Effective deblurring performance necessitates only the generator from the deblur learning module, akin to other models. However, what distinguishes the proposed motion deblurring network is its construction, which involves the connection of two modules characterized by distinct features. Furthermore, leveraging the directly captured UAV image dataset from bridge inspections for training offers the advantage of addressing the motion blur phenomenon frequently encountered in UAV environments. The proposed architecture, like other GAN-based deblur networks [22,23,29,30], consists of a generator and discriminator network.

### 2.2.2. Phase 1: Blur Learning Module

In the initial phase, denoted as Phase 1, a UAV dataset encompassing diverse pre-constructed scenes and high-clarity images is employed to introduce clear images into the generator. Additionally, to replicate various environments where blurring takes place, a noise map is integrated with the input image, capable of emulating a spectrum of conditions. Specifically, we sample a noise vector of length 4 from a normal distribution and employ it by replication across the spatial dimension. As depicted in Figure 3, the overarching generator architecture comprises two convolutional layers and nine residual block layers, with each residual block featuring an instance normalization layer and Relu activation. Within this process, the application of the global skip connection, as introduced in the deblurGAN model [22], is expected to yield advantages in terms of accelerated training and enhanced generalization outcomes.

The output generated through the training process constitutes a synthesized blurry image of the same dimensions as the input image. This synthesized image serves as the input to the discriminator network, which adopts the well-established VGG19 network architecture [25]. The ultimate output of this network is the probability assigned to classifying the synthesized blurry image as a genuine one. Both the generator and discriminator networks mentioned earlier undergo training with the aim of minimizing perceptual loss and adversarial loss, respectively. The perceptual loss is computed based on synthetic images generated by the generator network and images extracted from the UAV dataset. Meanwhile, the adversarial loss is calculated by comparing the generated blurred images from the newly created dataset with the actual blurred images.

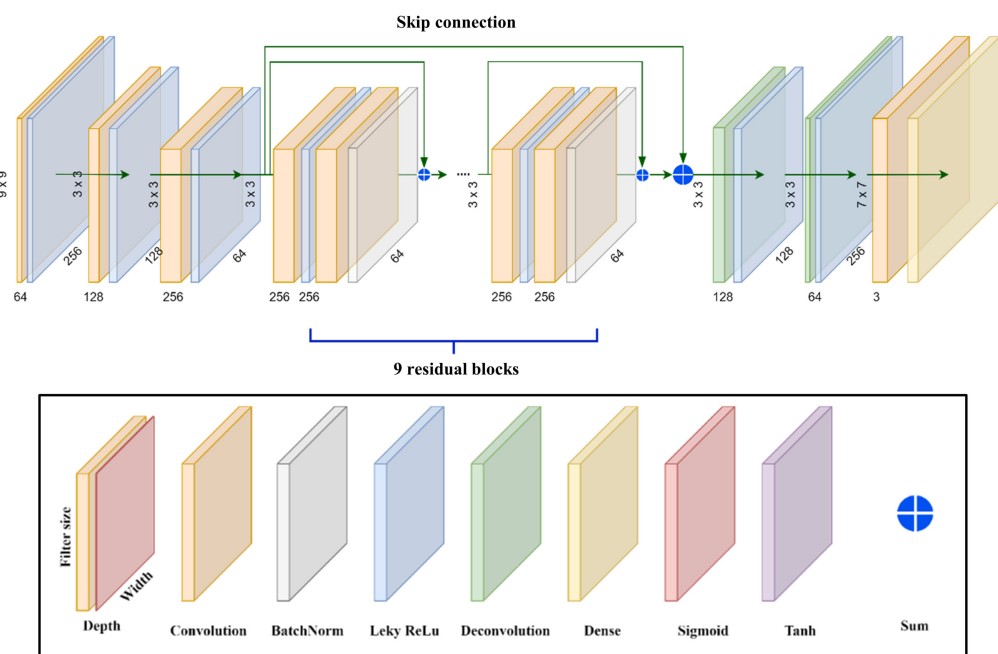

**Figure 3.** Generator of blur learning module.

### 2.2.3. Phase 2: Deblur Learning Module

In phase 2, a deblurring process is executed, relying on a synthesized blurry image that closely resembles a real one generated earlier. This synthesized blurry image serves as input to the generator network within the deblur learning module. Similar to the generator network employed in Phase 1, this architecture comprises a convolutional layer followed by an additional 16 residual block layers. The output of discriminator network provides the probability that the generated sharp image convincingly resembles a genuine sharp image, incorporating perceptual loss and adversarial loss as shown in Figure 4. All forms of loss functions are computed based on the generated sharp images, and the perceptual loss function is employed for updating the model.

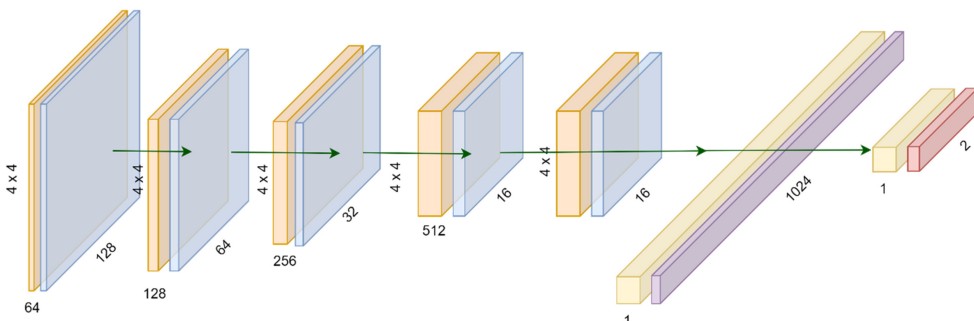

**Figure 4.** Discriminator of deblur learning module.

While the proposed approach employs a loss function concept similar to that proposed by Johnson et al. [31], it leverages features before the ReLU activation layer when computing the perceptual loss. Mean Squared Error (MSE) stands as one of the widely utilized loss functions in the field of image restoration. The MSE value is calculated to quantify the content loss between the original image and the generated image. We enhanced the performance of the generator network within the blur learning module by formulating a relative loss function aimed at generating synthesized blurred images that closely mimic

the characteristics of real blurred images. This loss function also facilitates continuous model refinement to deceive the discriminator network effectively.

### 2.2.4. Loss Functions

The deblur module is trained using perceptual loss and content loss, and additionally incorporates the concept of relativistic loss [32]. The idea for relativistic loss comes from extending the role of the generator in the blur learning module (a network that generates fake blurred images) to better align the generated fake blurred images to the real blurry images. In addition, the real blurry images should be updated to reduce their probability of being real. Most standard GANs with non-saturating loss function assume the cross-entropy loss as

$$
\begin{aligned}
f_1(D(x)) &= -log(D(x)) \\
f_2(D(x)) &= -log(1 - D(x))
\end{aligned}
\tag{2}
$$

where $f_1$, $f_2$ are scalar-to-scalar functions, and $D(x)$ represents the probability of the input image $x$ being a real image in discriminator [33] and can be denoted as $\mathrm{sigmoid}(C(x))$. Here, $C(x)$ is a non-transformed discriminator output called a critic. In standard GAN, the term critic $C(x)$ refers to how authentically the discriminator assesses the input data. In this context, positive values of $D(x)$ indicate that the data appear realistic, while negative values suggest that the data appear fake. According to the extended relativistic loss announced by Jolicoeur-Martineau [32], Equation (2) can be modified to $D(x_r, x_f) = \sigma(C(x_r) - C(x_f))$, which can be interpreted as the discriminator assessing whether the provided real data are more realistic that randomly sampled fake data. Therefore, by developing this relativistic loss, the adversarial loss $D$ is defined as follows:

$$
\begin{aligned}
D(I_{real}, I_{blurry}) &= \sigma(C(I_{real}) - C(I_{blurry})) \to 1 \\
D(I_{fake}, I_{blurry}) = D(G(I_{real}, I_{sharp})) &= \sigma(C(G(I_{real})) - C(G(I_{sharp}))) \to 0
\end{aligned}
\tag{3}
$$

where generator $G$ is trained to increase the probability that synthesized images are looks more realistic $(0 \to 1)$, and at the same time decrease the probability that real images look realistic $(1 \to 0)$.

By applying the concept of relativistic loss defined in this way to the blur learning module, Equation (2) contributes to fooling the $D$ into thinking that a fake (generated) blurry image is similar to the real blurry image $(0 \to 1)$, and at the same time, $G$ is trained to reduce the probability that a real blurry image is real $(1 \to 0)$.

Given by Equation (3), the relativistic loss in the blur learning module is defined as $B1$ loss:

$$
L_{B1} = -[\log((D(I_{real}, I_{blurry})) - D(G(I_{input}))) + \log(1 - (D(G(I_{input}) - D(I_{real}, I_{blurry}))))].
\tag{4}
$$

Based on the $B1$ loss, the relativistic loss of the generator in the deblur learning module is

$$
L_{B2} = -[\log((D(I_{real}, I_{sharp})) - D(G(I_{input}))) + \log(1 - (D(G(I_{input}) - D(I_{real}, I_{sharp}))))].
\tag{5}
$$

In the training phase, the loss functions of the blur and deblur learning modules are constructed by multiplying their respective weight values as follows:

$$
\mathcal{L}_{blurlearning} = L_{perceptual} + \beta \cdot L_{B1}, \mathcal{L}_{deblurlearning} = L_{perceptual} + \alpha \cdot L_{content} + \beta \cdot L_{B2}.
\tag{6}
$$

where $\alpha$ denotes the weighted parameter of content loss and $\beta$ denotes weighted parameter of $B1$ and $B2$ loss. The two-weighted parameters $\alpha$ and $\beta$ were inferred to be suitable values within a range that ensures the stability of the proposed model through multiple rounds of training and optimization, preventing the overfitting of specific loss components during the learning process.

## 3. Experiments and Discussions

### 3.1. Generation of the Image Dataset

The blur image dataset used in the blur learning network and the clear image dataset used in the deblur learning network learning consist of images taken of the sides and bottom of the bridge. The target structure is a bridge consisting of 10 steel boxes and 10 PSC box girders. In the case of the pier, the route was set at an interval between 2.42 and 2.49 m, and images were captured by automatic flight as shown in Figure 5. In addition, in the case of the deck plate, considering that it is a GPS-denied area, images were acquired at a distance of 4 to 5 m at a level where sufficient communication is possible. Some images of areas near the floor where GPS signals are weak were obtained by the pilot via manual flight and caused a number of blurs. The overall process was taken using UAVs in the same environment as normal bridge monitoring and a DSLR camera mounted on UAVs secured 1100 blur images and 2000 clear images from 20 different scenes. Among the bridge images acquired in the real world, 900 of the 1100 blurred images were used as training datasets, the remaining 100 as validation datasets, and 100 as test sets to learn the blur learning module. In addition, out of 2000 clear image data required for the deblur learning module, 1600 were used as training datasets, 200 were used as test sets and the remaining 200 were used as validation datasets. Each image is in JPEG format, compressed at 50% of the original size.

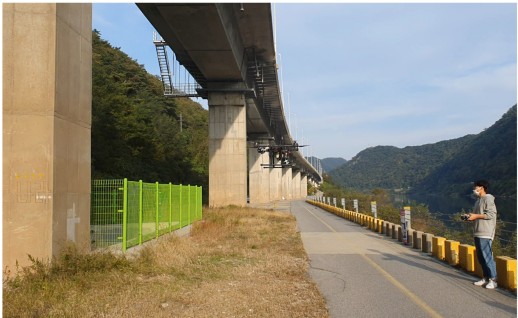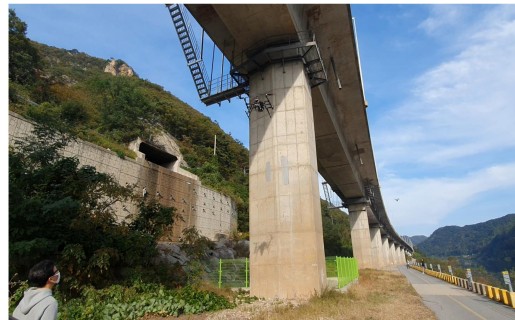

**Figure 5.** Image acquisition through autonomous–manual flight and target bridge.

A quadcopter-type UAV system was used to acquire the image dataset. The UAV system is equipped with a Mini-PC for real-time image storage and coordinate mapping, a Sony Alpha 7 or Alpha 9 DSLR camera, and a ZEISS Batis 85 mm f/1.8 lens, using a customized three-axis gimbal to reduce vibration as shown in the Figure 6. The gimbal used an iPower IR6212H-50T brush motor with a roll, pitch ±90° operating range and an additional IMU sensor for control of the gimbal was attached. With these equipment settings, stable image acquisition was performed with minimal probability of motion blur occurrence.

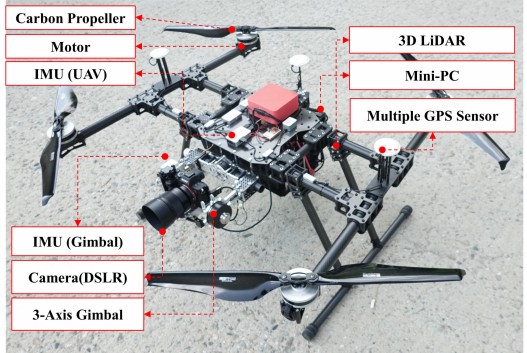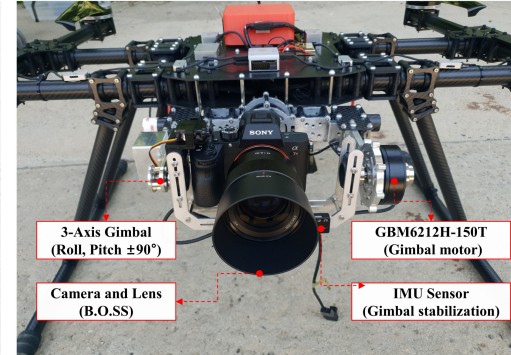

**Figure 6.** UAV systems and components for image acquisition.

The values of the camera's internal parameters were set differently depending on the region of interest. Since the illumination environment changes depending on the location of each member, the values of ISO sensitivity, property size, and shutter speed were adjusted accordingly as shown in Table 1.

**Table 1.** Camera parameter setting according to the amount of illumination used in this study.

| Sides of Bridge (Pier and Girder) | | Bottom of Bridge Deck | |
|---|---|---|---|
| ISO sensitivity | 100∼320 or Auto | ISO sensitivity | 320∼500 |
| Aperture size | f/4.5∼f/5.6 | Aperture size | f/1.8∼f/3.5 |
| Shutter speed | 1/250∼1/800 s | Shutter speed | 1/100∼1/250 s |

### 3.2. Training and Implementation Details

During the training process, image augmentation techniques were employed to introduce variations such as image rotation (−20° to +20°), horizontal and vertical movement, and horizontal and vertical flipping. This augmentation technique helps minimize overfitting and allows for training with a diverse range of data using a relatively small number of images captured by the UAV. Additionally, to simulate blur effects occurring in different environments, the dataset was composed by applying various illumination values (−30 to +30). In the training of the blur and deblur module, the weight initialization was carried out using a Gaussian distribution characterized by a mean of 0 and a standard deviation of 0.01. After processing a mini-batch consisting of four samples, the model's weights were updated in each iteration. To diversify the training dataset, a $128 \times 128$ patch was extracted from arbitrary positions within an image. To further enhance dataset diversity, frames were randomly flipped. The employed learning rate followed a scheme of annealing, commencing at $10^{-4}$ and progressively decreasing to $10^{-6}$ upon a convergence of the training loss. The weighted parameters $\alpha$ and $\beta$ were adjusted to 0.05 and 0.001, respectively.

### 3.3. Experimental Results and IQA

To validate the effectiveness of the proposed method, which constructs training data pairs using a blur module that mimics the blurring effects occurring in actual UAV to create blurry images, and trains a deblur module using these data, we applied it to real-world blurry images. Figure 7 shows a blurry image taken during a bridge inspection using a UAV and the deblurred image obtained by applying the proposed model. A total of 100 images were tested, using the actual raw data images ($6000 \times 4000$, 24 MP) compressed to 50% of their size. Blurred images were taken in environments where motion blur is likely to occur, such as under bridges with insufficient light, and only images with motion blur were used, excluding cases with out-of-focus blur. Before validating, we added the blurry images generated by the blur module to the training samples for training the deblur module in order to improve the performance of the deblur module. This method was effective in fine-tuning the deblur module, and the deblur performance improved compared to when this was not performed.

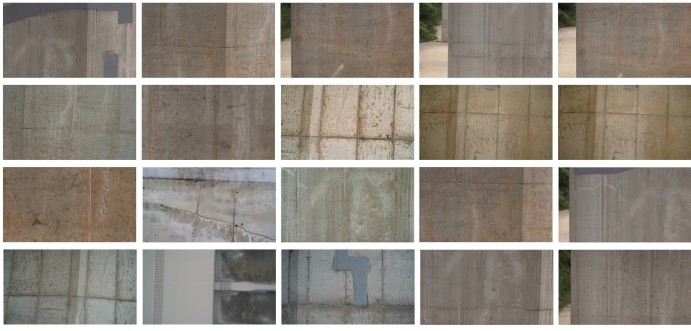

**Figure 7.** Representative various scenes of UAV image dataset.

Utilizing the results obtained from training the deblurring network to enhance given blurry images as shown in Figure 8, we conducted a quality evaluation using three widely used image quality metrics. The commercial metrics employed for image quality assessment included BRISQUE [34], NIQE [35], and LBMS (Local Blur Map Score) [36], which evaluates the quality of blur images based on local blur maps. Table 2 shows a comparison of the quality scores before and after performing dynamic deblurring and presents the average values obtained by applying the metrics to each of the 100 test images. When comparing quality using BRISQUE, the proposed multi-module-based deblurring model exhibited an approximate 36.6% enhancement in quality. Additionally, the NIQE evaluation results also indicated an improvement of about 37.68% in quality scores. Lastly, the LBMS-based evaluation of blur image quality demonstrated an enhancement effect of approximately 33.99%, affirming that the motion deblurring network employed earlier led to improved image quality.

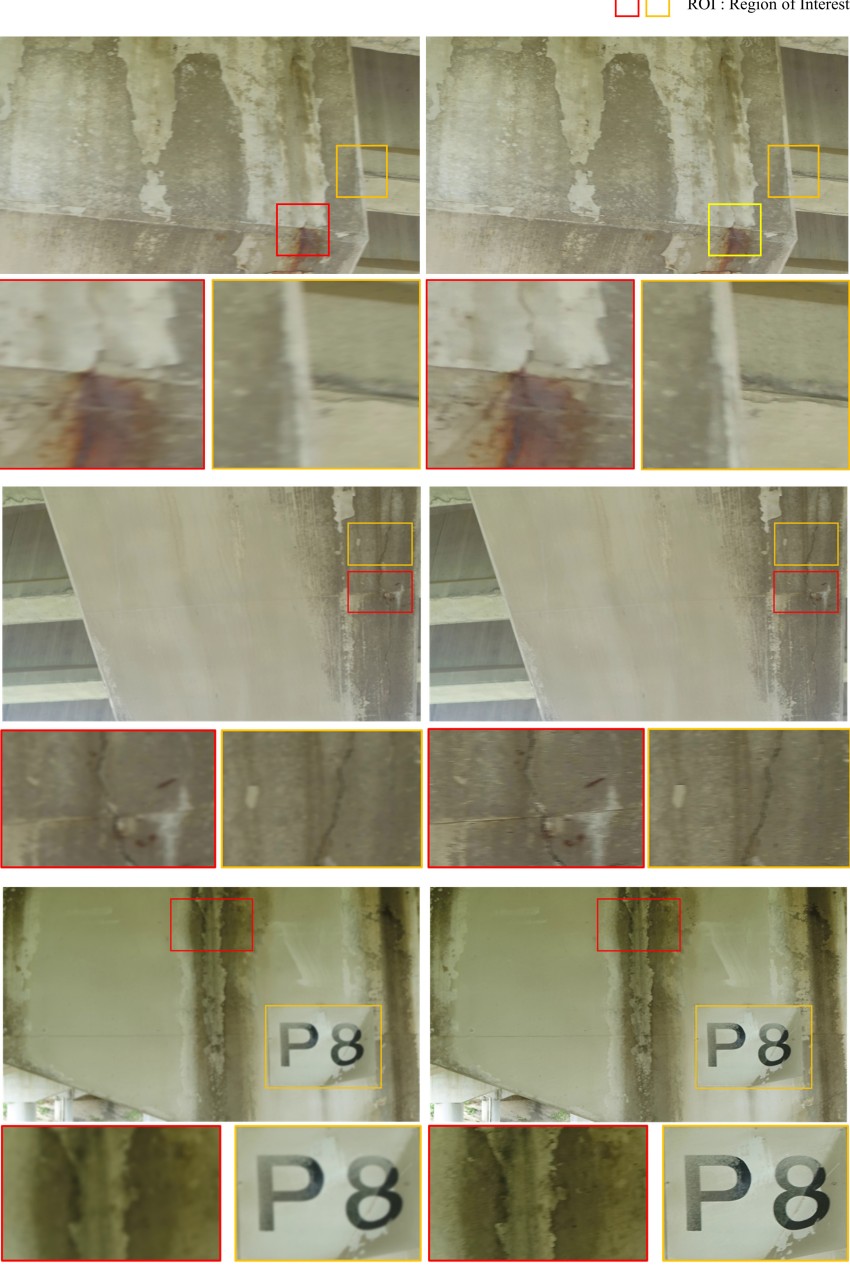

**Figure 8.** Performing deblurring on blurring images between bridge inspections using UAV (**left**) blurred image, (**right**) deblurred image.

**Table 2.** Comparison of image quality evaluation results before and after deblurring.

| Measure | Average Score | | |
| --- | --- | --- | --- |
| | **Blurry Images** | **Deblurred Images** | **Changes** |
| BIRSQUE | 35.94 | 22.77 | +36.6% |
| NIQUE | 6.21 | 3.87 | +37.68% |
| LBM | 0.606 | 0.812 | +33.99% |

## 4. Validation of Deblur Effect in Object Detection

An experiment was designed to verify the impact of the proposed motion deblur network's output on object detection. Specifically, the detection of cracks holds high importance in the process of infrastructure inspection using UAV. Therefore, the experiment compared the results of crack detection using blurry images and deblurred images directly. The crack detection process was simulated using actual images acquired through UAVs. The ultimate goal was to compare the detection results and analyze the effects of image deblurring.

### 4.1. Deep Learning Model for Detecting Object

The YOLO (You Only Look Once) network includes single-stage object detectors. In this deep learning model, image frames are featurized through a backbone. These features are combined and fused in the intermediary section, and subsequently forwarded to the network's head. YOLO then predicts the positions and categories of objects for which bounding boxes need to be delineated. It achieves efficient inference speed by employing extended efficient layer aggregation. Furthermore, it enhances the model architecture by establishing connections between layers and simultaneously adjusting the network's depth and breadth. Moreover, the YOLO model employs a re-parameterization strategy to identify which network modules should integrate this approach, guaranteeing the smooth flow of gradient propagation. These adjustments together result in enhanced performance, both in terms of speed and accuracy, for tasks related to object detection. In this study, the performance of the proposed GAN-based deblur network is evaluated by comparing the object detection results using both the blurry images and the deblurred images with cracks. A dataset consisting of numerous UAV-acquired images containing cracks was created for training the object detection model. As shown in Table 3, the YOLO-v7 model [37] pretrained with the MS COCO dataset [38] has an AP of 51.40% and a Batch 32 Average time of 2.8 ms for 640 × 640 images. The YOLO-v7 network used in the validation experiment is a pre-trained model designed for 640 × 640 image size, and the training dataset was also resized to match the dimensions of 640 × 640. By applying the blur images and their corresponding deblur images to the pre-trained model, the object detection results can be compared to assess the improvement achieved through the deblurring process. All training processes were executed in the Google Colab environment, utilizing an Intel Xeon CPU@2.20 GHz, 83.5 GB of RAM, and an NVIDIA A100-SXM-40 GB GPU.

**Table 3.** The performance of YOLO-v7 series (MS COCO dataset) [37,38].

| Model | Test Size | AP (Test) | AP (50, Test) | Batch 1 fps | Batch 32 Average Time |
| --- | --- | --- | --- | --- | --- |
| YOLO-v7 | 640 | 51.40% | 69.70% | 161 fps | 2.8 ms |
| YOLO-v7(X) | 640 | 53.10% | 71.20% | 114 fps | 4.3 ms |
| YOLO-v7(W6) | 1280 | 54.90% | 72.60% | 84 fps | 7.6 ms |

### 4.2. Image Dataset for Training Model

In order to evaluate the performance of the proposed image quality improvement (motion deblurring) method, a validation was conducted to compare the object detection results using the YOLO-v7 network. The image dataset used in this study comprised a total of 10,900 crack images (CrackNet dataset [39]) as shown in Figure 9. Among

these, 9900 images were used for training, and 1000 images were reserved for validation. Image data augmentation techniques were applied, including vertical and horizontal flips, rotation within the range of −15 degrees to +15 degrees, and brightness adjustments ranging from −25% to +25%. Additionally, noise and blur data were added to enhance the object detection performance for blurred images. The weight model utilized transfer learning, using the pre-trained segmentation model. Each image was assigned four types of labels: slanted, crack, horizontal crack, and vertical crack. Both the image data and the corresponding label were set as input values for the training course, and the training was carried out through a total of 25 epochs.

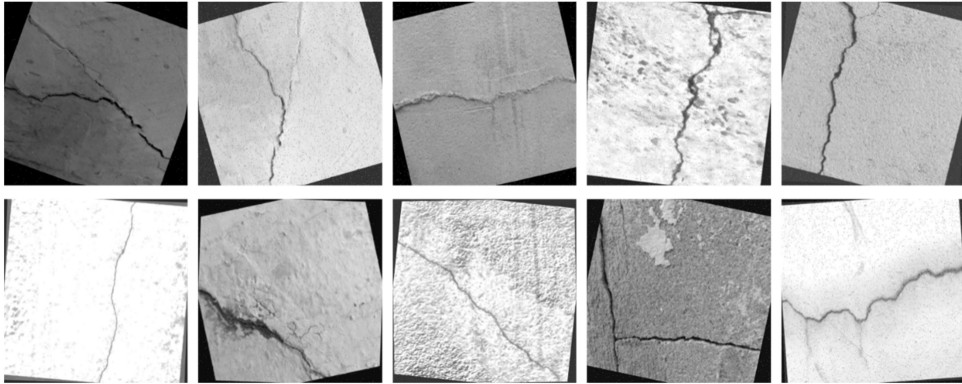

**Figure 9.** Representative results of image augmentation.

### 4.3. Training and Implementation Details

This study attempted to increase detection performance and reduce learning time through transfer learning using pre-trained models [40]. Pre-trained models have a weighting scheme created by learning the characteristics of various images. If transfer learning is not conducted, deep learning weights begin learning from random initial values, but the learning effect is further improved by using pre-trained weights through such transfer learning. All YOLO-v7 networks used in the validation experiment used pre-trained weights as initial values of COCO dataset. The hyperparameter of the deep learning model is a condition that defines the network structure and driving environment, and hyperparameter optimization is the most important factor that determines the performance and learning time of the model. Adjusting the weight by injecting all the training data corresponds to one epoch, and the verification score is calculated for each epoch. After finishing the entire epoch, we selected the one model with the best performance. Reflecting these verification results, follow-up tasks, such as adjusting the hyperparameters to improve generalization performance, were performed. Once the training and validation processes were completed, the performance of the models was evaluated through a test evaluation. The model with the highest mean Average Precision (mAP) value at an IoU threshold of 0.5 was selected. Precision, recall, and mAP@0.5 were applied as performance metrics for object detection.

### 4.4. Validation Results

Tables 4 and 5 present the test evaluation results for the YOLO-v7 segmentation model, including precision, recall, and mAP@0.5 for both bounding box and mask predictions. Table 6 shows the confusion matrix for the YOLO-v7 segmentation model's test results. To assess the overall object detection performance, mAP@0.5 was calculated using the area under the Precision–Recall (PR) curve as depicted in Figure 10. In terms of bounding box predictions, the crack class achieved the highest mAP@0.5 value of 0.765. Regarding mask predictions, the highest mAP@0.5 value of 0.642 was observed in the horizontal crack class. In addition, as shown in Figure 11, the F1-score also showed more accurate prediction and reproduction than expected with values of 0.66, 0.62 in each bounding box and mask segmentation. The detection results for cracks were favorable, as the characteristics of these

objects were distinct. Notably, the detection accuracy for vertical and horizontal cracks was relatively higher compared to slanted cracks. It is challenging to generalize the crack detection performance of the YOLO-v7 network due to the significantly higher number of crack instances compared to the other three classes.

**Table 4.** Test performance of YOLO-v7 segmentation (bounding box).

| Class | Number of Instances | Precision | Recall | mAP@0.5 | mAP@0.5:0.95 |
|---|---|---|---|---|---|
| Crack | 990 | 0.74 | 0.774 | 0.765 | 0.619 |
| Horizontal Crack | 132 | 0.715 | 0.646 | 0.693 | 0.533 |
| Vertical Crack | 131 | 0.705 | 0.557 | 0.613 | 0.477 |
| Slanted Crack | 116 | 0.674 | 0.5 | 0.547 | 0.478 |
| All | 1369 | 0.709 | 0.619 | 0.654 | 0.527 |

**Table 5.** Test performance of YOLO-v7 segmentation (mask, segmentation).

| Class | Number of Instances | Precision | Recall | mAP@0.5 | mAP@0.5:0.95 |
|---|---|---|---|---|---|
| Crack | 990 | 0.626 | 0.655 | 0.561 | 0.171 |
| Horizontal Crack | 132 | 0.681 | 0.616 | 0.642 | 0.245 |
| Vertical Crack | 131 | 0.695 | 0.55 | 0.595 | 0.224 |
| Slanted Crack | 116 | 0.673 | 0.5 | 0.539 | 0.203 |
| All | 1369 | 0.669 | 0.58 | 0.584 | 0.211 |

**Table 6.** Confusion matrix of YOLO-v7 segmentation.

| Prediction/Truth | Crack | Horizontal Crack | Vertical Crack | Slanted Crack |
|---|---|---|---|---|
| Crack | 0.78 | 0.26 | 0.33 | 0.33 |
| Horizontal crack | 0.01 | 0.63 | 0 | 0.02 |
| Vertical crack | 0 | 0 | 0.55 | 0.09 |
| Slanted crack | 0.01 | 0.01 | 0.02 | 0.5 |

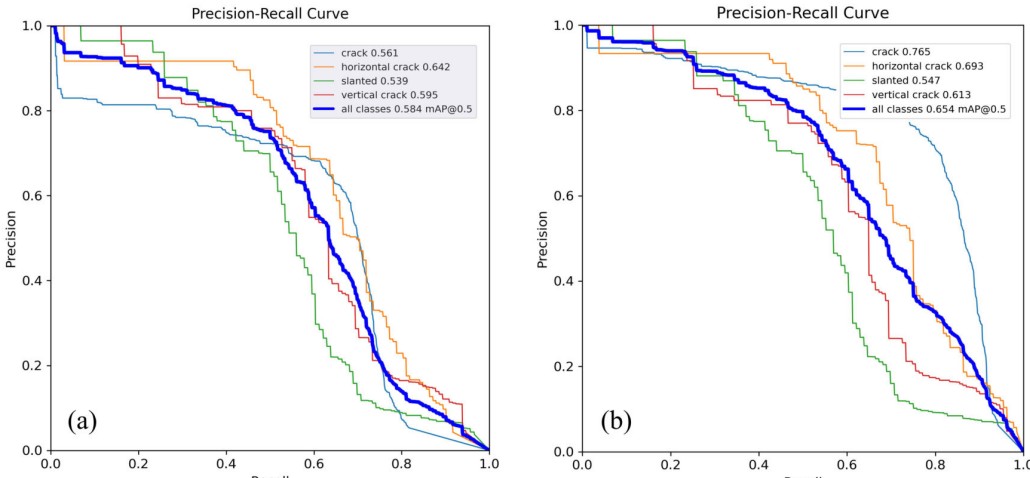

**Figure 10.** The results of the Precision–Recall (PR) curve for (**a**) box bounding and (**b**) mask segmentation obtained through learning.

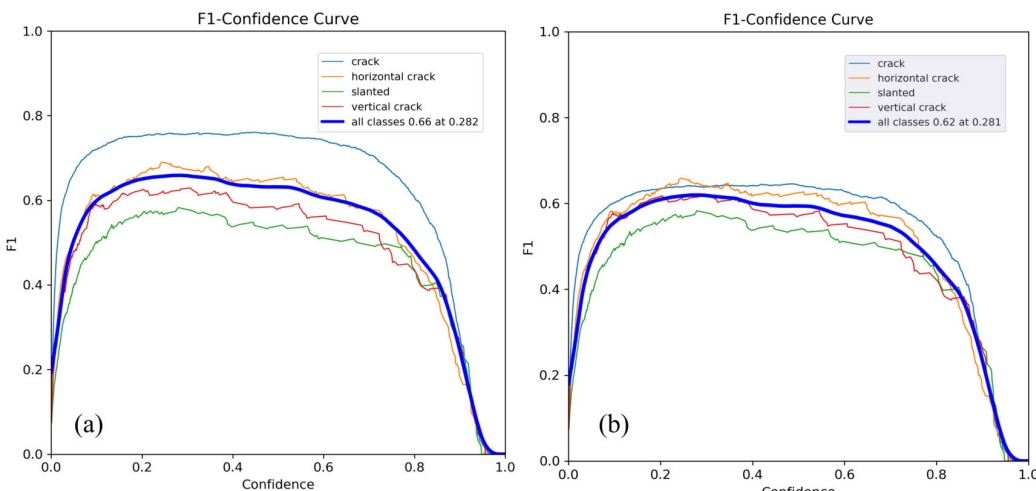

**Figure 11.** The results of the F1 score curve for (**a**) box bounding and (**b**) mask segmentation obtained through learning.

### 4.5. Test Results

Using the trained and validated crack detection model, tests were conducted on a set of 19 blurry images and another set of 19 deblurred images. The original image size was $3840 \times 2160$, and for consistency with the training image size of the YOLO-v7, the images were cropped to $640 \times 640$ before performing object detection. The original set of 19 images and the deblur image set contain the same region of interest (ROI), and visual inspection in the field shows that a total of 12 images contain cracks, and 7 images do not contain cracks as shown in Figure 12.

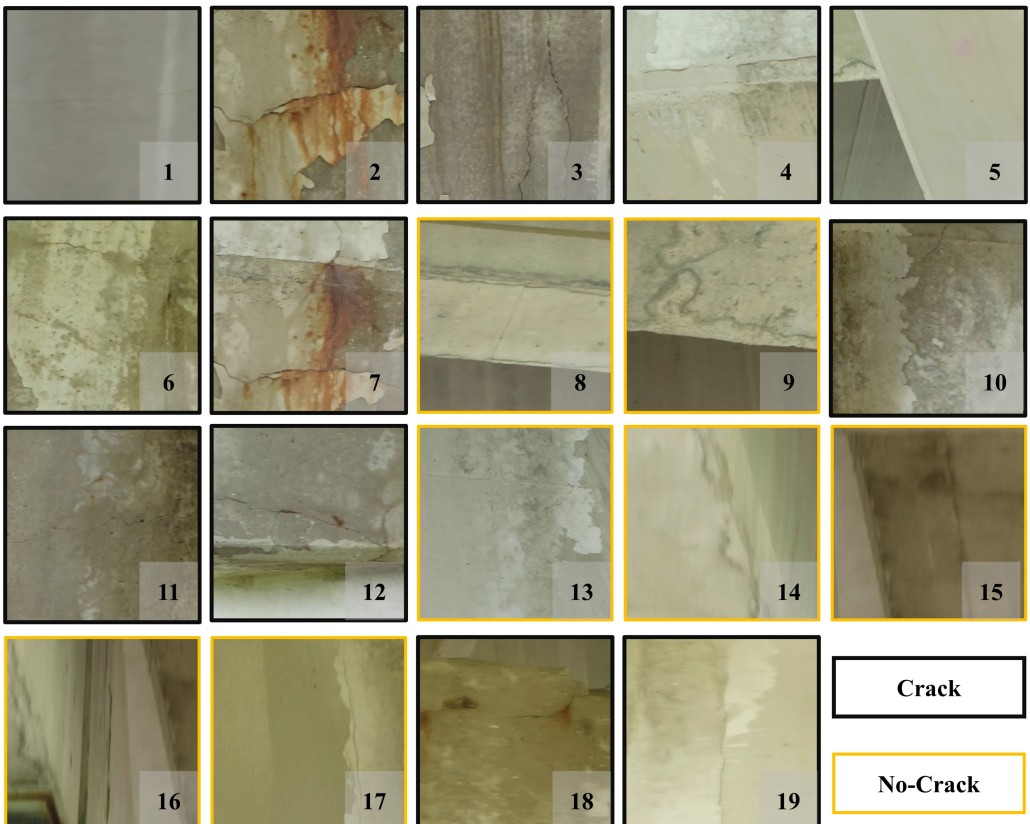

**Figure 12.** Test image set for crack detection (original and deblurred).

Generally, four main metrics are commonly used to evaluate the performance of an object detection model. Precision, which measures the precision rate, calculates the ratio of correctly predicted positive instances to all instances predicted as positive. In other words, it focuses on the accuracy of positive predictions. Recall, which indicates sensitivity, measures the ratio of correctly predicted positive instances to all actual positive instances. Accuracy, which represents the overall accuracy of the model predictions, calculates the ratio of correct predictions (both true positives and true negatives), to all instances. Lastly, the F1 score is calculated as the harmonic mean of precision and recall.

The results of crack detection using the original and deblurred image sets are shown in Figure 13. When applying crack detection to the original image set, out of 19 images (23 instances), 5 images (6 instances) were classified as true positive, and 4 images (10 instances) were classified as false negative. There were no false positive and 7 true negatives. Based on these classifications, the four evaluation metrics were calculated and are presented in Table 7. Using the same object detection model, the results for the original image set as input showed P = 100%, R = 37.50%, Accuracy = 58.33%, and F1 score = 54.54. When using the deblurred image dataset as input, which was obtained by applying the deblurring network to the original images, there was a significant improvement in the performance. The accuracy increased by 16.67% to 75.00%, recall improved by 27.2% to 64.70%, and the F1 score increased by 24.02% to 78.56.

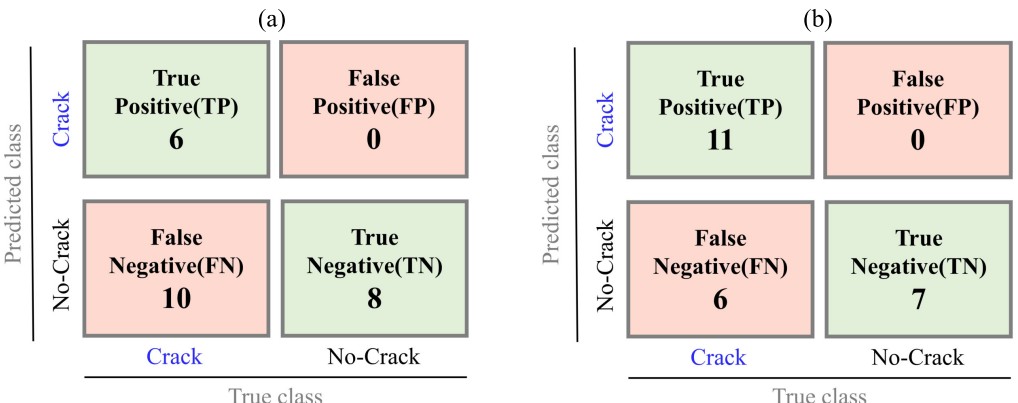

**Figure 13.** Crack detection performance comparison results. (**a**) Original. (**b**) Deblurred.

**Table 7.** The result of YOLO-v7 for detecting cracks (confidence = 0.25).

| Evaluation Metric | Original | Deblurred | Change |
|---|---|---|---|
| Accuracy (%) | 58.33 | 75.00 | +16.67 |
| Precision (%) | 100 | 100 | 0 |
| Recall (%) | 37.50 | 64.70 | +27.2 |
| F1 score | 54.54 | 78.56 | +24.02 |

Figure 14 shows the representative results of object detection performed using the original image captured by a UAV and the improved quality image after deblurring. Each image is annotated with numbers corresponding to visually identifiable instances, and instances identified as cracks are illustrated with segmentation masks and bounding boxes. The results of deblurring indicate that the progression of cracks within the images was recreated in a clear state, regardless of their orientation, under motion blur conditions. For a more quantitative comparison, Table 8 shows the classification results of crack detection among 9 out of 12 images, including instances. Furthermore, upon analyzing the crack detection images from both datasets, as shown in Figure 14, it can be observed that the crack detection rate (class probability) is higher in the deblurred images compared to the blurry images with poor image quality (#1, #3, #19). It is also evident that cracks were detected in 4 images that were previously undetected prior to deblurring. However, in

two instances (#4 and #12), false alarms occurred, leading to a decreased object detection rate after deblurring as shown in Table 7. In these cases, the bounding box detection failed to accurately identify the class, while the mask segmentation displayed a higher value. This suggests that these false negatives can be attributed to the learning process of the YOLO model rather than being a result of the deblurring network's performance. Consequently, based on these detection results, it can be inferred that the image deblurring effect achieved using the multi-module-based generative model enhances the accuracy of damage detection to a certain extent.

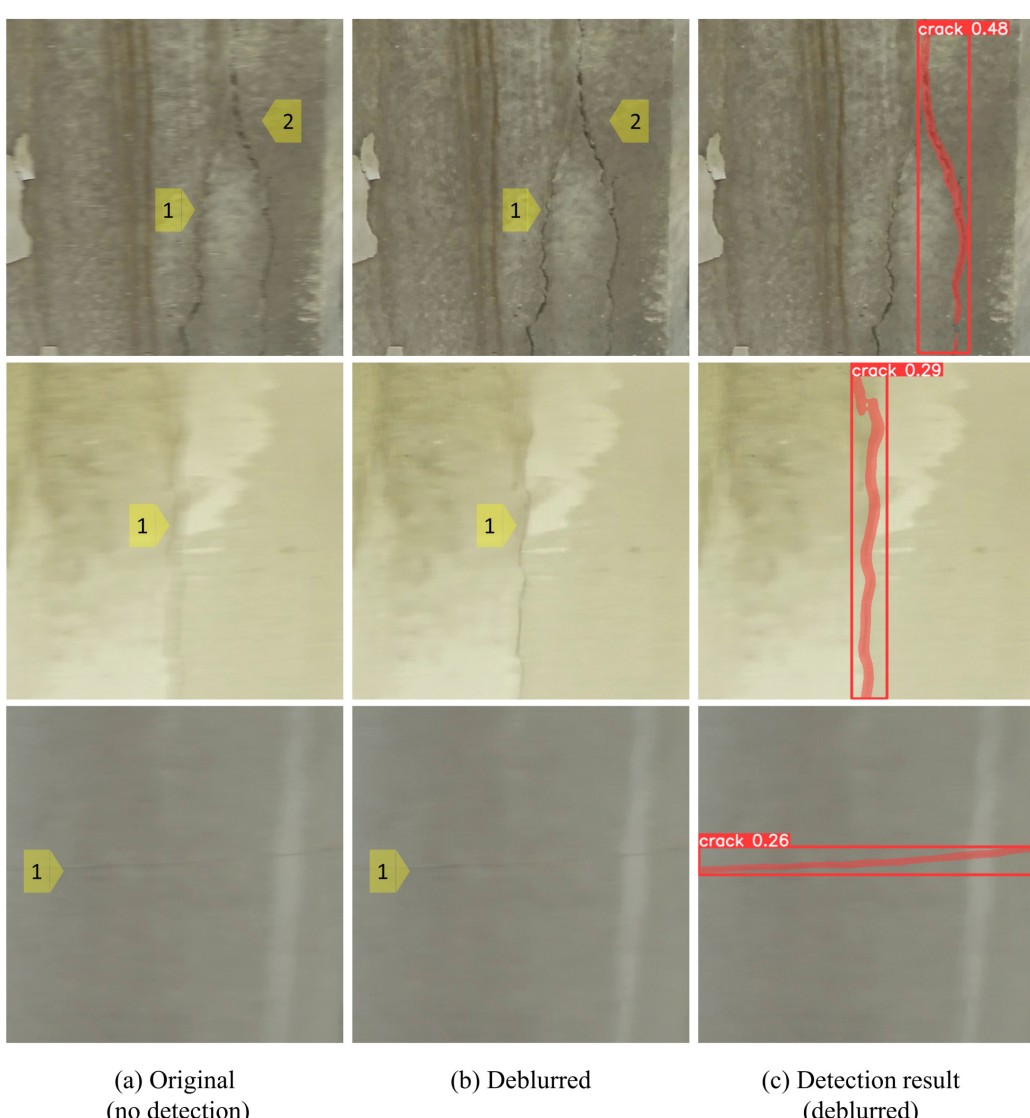

|  (a) Original | (b) Deblurred | (c) Detection result |
| (no detection) | | (deblurred) |

**Figure 14.** Representative result of object detection after deblurring. (#1, #3, #19).

**Table 8.** The result of object detection using YOLO-v7.

| Number of Image | Original Images | | Deblurred Images | | |
|---|---|---|---|---|---|
| | Class | Probability | Class | Probability | Changes |
| 1 | No detect | No detect | 1 crack | 0.26 | ▲ 0.26 |
| 2 | 1 crack | 0.3 | 2 crack | 0.55 | ▲ 0.25 |
| 3 | No detect | No detect | 1 crack | 0.46 | ▲ 0.46 |
| 4 | 2 crack | 0.70, 0.3 | 2 crack | 0.43 0.32 | ▽ 0.27 ▲ 0.02 |
| 5 | 1 crack | 0.26 | 1 crack | 0.37 | ▲ 0.11 |
| 7 | No detect | No detect | 3 crack | 0.44 | ▲ 0.44 |
| 11 | 1 crack | 0.29 | 1 crack | 0.36 | ▲ 0.07 |
| 12 | 1 crack | 0.37 | 1 crack | 0.33 | ▽ 0.34 |
| 19 | No detect | No detect | 1 crack | 0.29 | ▲ 0.29 |

## 5. Conclusions

In the UAV-based bridge inspection technology, the occurrence of motion blur in images poses a significant threat to the reliability of inspection results. Therefore, this study introduces a standard GAN-based deblurring network model aimed at mitigating and enhancing the quality of images affected by motion blur during UAV-based bridge inspections. Unlike conventional deblurring networks based on GANs that utilize artificially generated images through simulations without accounting for the distinct characteristics of blur during the learning process, our proposed model advances by generating synthetic images that closely emulate real-world motion blur traits. Consequently, we successfully acquired a collection of blurred images resembling various scenarios encountered in UAV-based imaging, which serve as the foundation for subsequent deblurring processes.

Furthermore, validation experiments using three widely used image quality metrics demonstrated an improvement in image quality of approximately 33% to 36%. Additionally, in the context of crack damage detection using the latest object detection model, we conducted validation experiments that showed higher instance detection rates and accuracy in deblurred images compared to the original images. These experiments involved training the YOLO-v7 deep learning model, and the results indicated a 16.67% increase in instance detection accuracy, a 27.2% improvement in recall, and a 24% increase in F1 score. While two instances of false alarms occurred during the validation process, it was concluded that these were not due to constraints in the learning of the deblurring model but rather performance limitations of the object detection model.

Therefore, based on these validation results, the motion deblurring network proposed in this study are expected to enhance the reliability and accuracy of bridge inspection technology using UAVs.

**Author Contributions:** Conceptualization, J.-H.L., I.-H.K. and H.-J.J.; methodology, J.-H.L.; software, J.-H.L.; validation, I.-H.K. and J.-H.L.; formal analysis, G.-H.G. and J.-H.L.; investigation, G.-H.G. and J.-H.L.; data acquisition, G.-H.G. and J.-H.L.; writing—original draft preparation, J.-H.L. and I.-H.K.; writing—review and editing, I.-H.K. and H.-J.J.; visualization, J.-H.L. and I.-H.K.; supervision, H.-J.J.; project administration, H.-J.J.; funding acquisition, H.-J.J. All authors have read and agreed to the published version of the manuscript.

**Funding:** This study is supported by the National Research Foundation of Korea (NRF) grant funded by the Ministry of Science and ICT (NRF 2017R1A5A1014883) and 'Research Center for Smart Submerged Floating Tunnel System' and by the Korea Agency for Infrastructure Technology Advancement (KAIA) grant funded by the Ministry of Land, Infrastructure and Transport (Grant RS-2020-KA156208).

**Data Availability Statement:** Not applicable.

**Conflicts of Interest:** The authors declare no conflict of interest.

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
