# Peer review of "A Motion Deblurring Network for Enhancing UAV Image Quality in Bridge Inspection"

_drones, doi:10.3390/drones7110657_

Round 1

Reviewer 1 Report

Comments and Suggestions for Authors

The manuscript describes a method for image processing to remove blur. The authors strongly focus on photos used to monitor the technical condition of the bridge. At the same time, they do not indicate the universality of the method being developed.

Regarding the introduction and background of the research: the issue of dealing with blur in photos was described too generally, please refer to the technical issues of solving this problem at the level of photo acquisition in the case of drone flights in the vicinity of bridges.

Please describe in more detail the issue of bridge monitoring based on drone photos using the described image processing technology (how does the described method relate to the assessment of bridge damage by the drone operator/accompanying person in real time?).

Please correct figure no. 3, some descriptions have too small a font (lower part of the figure).

Please describe the training set in more detail.

Reviewer 2 Report

Comments and Suggestions for Authors

Dear authors,

Thanks a lot for your work. Very interesting indeed. I like very much the strategy of blurring and deblurring strategy. It seems to be consistent with the reported results as well with the implementation of using YOLO techniques.

In the electrical domain, we also intentionally add noise to our signals, and I am wondering if the selected contamination does not influence the output. It means you reported that image rotation and illumination were the fundamental variables to create blurring.

But have you tried for example shooting with lower speeds?  

One of the typical blurring situations by flying drones and acquiring images is the misalignment of the speed and movement artifacts of the UAV platform (such as wind, stability of the platform, and vibrations) and the shooter speed of the camera. You did not report the program that you set up in your camera (section 3.1) (eg. automatic, aperture, speed, manual, etc. in line with the variables reported in lines 59 and 60). The same applies to the selected format (RAW vs. jpeg. In line 322 you mentioned compressed, does it mean jpeg format?)

In addition, it could be interesting to have more details about your gimbal. Because they are intended to minimize the artifacts associated with movements, resulting in the stabilization of your camera.

Have you tried as well to create oblique images? Depending on the field of view of your camera and the incident light, you are able to detect cracks. For example, Figure 12, row C, can substantially improve if you change the position of your platform.

It could be advantageous, to shoot at different times during the day.

Have you executed a photogrammetry method, in parallel to your study using the same images? Or does your UAV mission consist of positioning the platform at a certain point and acquiring the images?

It could be something that you can also mention in your document.

I also suggest making available your dataset to the scientific community.

In the Figure 2, I didn’t understand the G and D blocks. Maybe I got confused, sorry.

I guess your C(x) variable is not properly reported in equation (2). Same paragraph, I suggest changing Alexia by Jolicoeur-Martineau [32]. Alexia is the first name of the author.

On line 295, you mentioned the “hyperparameters” but you have not introduced such concepts. Moving lines 394 to 397 shortly before this line may clarify such concepts. What does mean alpha and betta?

I suggest citing the CrackNet database properly, as well as COCO.

To conclude, what is the strategy to select your sub-sample of 640x640 image from your original photos (6000x4000)?

Have you considered extending your study to different scenarios such as roads, vertical walls, and buildings?

Thanks a lot, really nice work.

Reviewer 3 Report

Comments and Suggestions for Authors

I have the following concerns

1. Additive model(1) is a partial case. How will your method work for multiplicative model.

2. It is known that GANs have been used for image processing for a long time. Specify what your difference is.

3. Why are vertical and horizontal cracks shown in tables 3, 4, 5. After all, they can be placed at an arbitrary angle to the drone. The question arises how to eliminate the dependence on affine transformations.

4. It is necessary to show how the accuracy depends on the level of blurring, and the F1-score depends on the level of illumination.

5. What was the ratio between training, test and validation samples.

6. References should be supplemented with articles for 2022-2023 to confirm the relevance of the presented research.

Comments on the Quality of English Language

Minor editing of English language required

Round 2

Reviewer 1 Report

Comments and Suggestions for Authors

I would like to thank the authors for their work and for improving the manuscript. The authors responded to my comments in detail. After the corrections, I am of the opinion that the article can be published in this form.

Reviewer 3 Report

Comments and Suggestions for Authors

I'm pretty much satisfied with the answers to my concern except 4.

Comments on the Quality of English Language

 Minor editing of English language required